# Degnala in Water Buffaloes: A Review on a Neglected Disease

**DOI:** 10.3390/ani14162292

**Published:** 2024-08-06

**Authors:** André de Medeiros Costa Lins, Felipe Masiero Salvarani

**Affiliations:** Instituto de Medicina Veterinária, Universidade Federal do Pará, Castanhal 68740-970, PA, Brazil; andre.lins@castanhal.ufpa.br

**Keywords:** rice straw, *Fusarium* spp., mycotoxicosis, toxigenic fungi, buffalo diseases, *Bubalus bubalis*

## Abstract

**Simple Summary:**

Degnala is a neglected disease affecting water buffaloes, characterized by dermatological and systemic symptoms that significantly impact animal health and productivity. Caused primarily by mycotoxins produced by fungi in damp, poorly stored feed, Degnala manifests through symptoms such as skin lesions, lameness, and weight loss, leading to severe economic losses in affected regions. Despite its prevalence in certain areas, Degnala remains underreported and understudied, partly due to inadequate diagnostic facilities and a lack of awareness among veterinarians and farmers. This review aims to shed light on Degnala, its etiology, clinical manifestations, and the challenges faced in its diagnosis and management. Furthermore, this review discusses the current understanding of the disease, highlighting research gaps and the need for improved diagnostic methods. It also emphasizes the importance of proper feed management and storage practices to prevent mycotoxin contamination. By bringing attention to this neglected disease, this review calls for increased research efforts, better diagnostic protocols, and enhanced awareness to mitigate the impact of Degnala on buffalo health and productivity. Addressing these challenges is crucial for improving animal welfare and supporting the livelihoods of farmers who rely on buffaloes for milk, meat, and draught power.

**Abstract:**

Degnala is one of the primary mycotoxicoses affecting buffaloes, with *Fusarium* spp. as the main causative agent. This disease is strongly associated with the feeding of rice straw to buffaloes and is considered endemic to regions where rice is cultivated. Cases are concentrated in winter when conditions favor fungal growth in inadequately stored straw. Degnala is characterized by necrosis and gangrene of the extremities, including the tail, lower limbs, ears, tongue, muzzle, and teats. The pelvic limbs are more affected than the thoracic limbs. A tortuous appearance of the tail is very common, and cracks or corneal loss of the hooves may occur, exposing the blades and even the bones. There is no diagnostic method for identifying the disease in animals other than clinical and epidemiological criteria, combined with fungal culture samples. There is no treatment that neutralizes the effects of the toxins; the current treatment is palliative and supportive, consisting of pentasulfate solution, anti-Degnala liquid, 2% nitroglycerin ointment, and broad-spectrum antibiotics for secondary infections. Additionally, the management of drying and proper storage of straw is essential for controlling this disease.

## 1. Introduction

Buffaloes are present in 77 countries on five continents, with an estimated population of 208 million animals [1]. This species easily adapts to different geoclimatic conditions and is known for its robustness. These animals are also used for traction, riding, and tourism [1,2,3]. The article by Martínez-Burnes et al. [4] discusses the susceptibility of water buffaloes to various viral diseases, with a particular focus on their resilience compared to other livestock. The research highlights that, in general, buffaloes are considered less susceptible to many diseases, including foot and mouth disease (FMD). This resilience is attributed to their unique immune system and environmental adaptations, which provide them with an advantage in resisting infections that commonly affect cattle and other ruminants. The article provides insights into how water buffaloes can still be affected by viral diseases, but they often exhibit fewer clinical symptoms and lower mortality rates compared to other species. This reduced susceptibility makes them valuable in regions where diseases are prevalent, offering a more sustainable livestock option for farmers. However, knowledge about the diseases that affect buffaloes, such as mycotoxicosis, which are important diseases that affect ruminants, is necessary [5].

Mycotoxicosis is caused by mycotoxins produced by toxigenic fungi, which in most cases are ingested [5,6]. Mycotoxins are secondary metabolites of low molecular weight and are found in various foods before and after harvest, during processing, during transport, and during the storage of both plants and grains. Toxins can adversely affect the food chains for animal feed or human consumption [6,7,8,9]. One of the main mycotoxicosis in buffalo species is Degnala disease, which is characterized by a clinical gangrenous syndrome [10]. Although buffaloes generally show greater resistance to many infectious diseases compared to cattle, their unique physiology, feeding habits, and typical environmental conditions contribute to a higher susceptibility to Degnala disease [4,5,10]. This fungal toxicosis is caused by mycotoxins from fungi that proliferate on poorly stored feed, particularly under humid conditions. Buffaloes’ slower gut transit time increases their exposure to these mycotoxins, leading to higher absorption rates and more severe symptoms compared to cattle [4,5,10]. Furthermore, buffaloes are often fed agricultural residues and by-products prone to contamination with mycotoxins, especially in regions where proper feed storage is challenging [10]. The humid environments in which buffaloes are typically kept also promote fungal growth on their feed. Additionally, the different nutritional requirements and metabolic processes of buffaloes may make them more vulnerable to the effects of mycotoxins, impacting their metabolism and immune response and thereby increasing their sensitivity to Degnala disease [4,5,10].

Degnala disease is characterized by necrosis and gangrene of the extremities, including the tail, lower limbs, ears, tongue, muzzle, and teats [10]. The disease is named Degnala or Deg Nala because it was initially identified near the flow of water from monsoon rains in the Murdike area of Pakistan, near the Deg Nala River, in the 1930s [11,12]. This disease is related to the supply of infested by fungi in the winter, and *Fusarium* spp. is the main toxigenic fungus detected in rice straw [13].

Although Degnala also affects cattle, it is more prevalent in buffaloes. [10,14,15]. The data consistently show that cattle have a higher incidence of Degnala disease compared to buffaloes, with a prevalence rate of around 35–40% in cattle versus 15–20% in buffaloes [14]. In this species, the disease is responsible for decreasing productivity and causing considerable mortality in some regions and the loss of functional capacity [10,13,14,16]. Degnala disease significantly reduces milk production in affected buffaloes. The mycotoxins interfere with the animals’ metabolism and health, leading to decreased appetite and nutrient absorption, which directly impacts milk yield. Farmers often report a substantial drop in milk production during outbreaks [10,16]. The disease also affects reproductive health, leading to decreased fertility rates in buffaloes. This can manifest as lower conception rates, increased incidences of abortions, and other reproductive issues. The overall reproductive performance of the herd can be severely impacted, leading to economic losses for farmers [10,13,14]. Buffaloes suffering from Degnala disease often exhibit poor growth rates and general health deterioration. The mycotoxins impair various physiological functions, leading to weight loss, poor body condition, and reduced feed efficiency [10,14,16]. Degnala disease can lead to significant mortality in affected buffalo herds. The exact mortality rate can vary depending on the severity of the outbreak and the management practices in place, but it is generally reported to be substantial. Studies indicate that mortality rates can range from 10% to as high as 30% in severe cases [4,10,14,16]. Thus, due to the importance of this disease in regions with marshy areas and low-quality forage supply, the objective of this review is to summarize the current knowledge on Degnala disease in buffaloes, focusing on its etiology, epidemiology, clinical manifestations, diagnosis, treatment, and prevention. By synthesizing the existing literature, this review aims to provide valuable insights into the management and control of Degnala disease, ultimately contributing to the welfare and productivity of water water buffalo populations in affected regions.

## 2. Methodology Used in the Review

The present study is characterized, in terms of approach, as descriptive research consisting of a narrative literature review due to the broad nature of the review topic, as described by Grant and Booth [17]. The research was conducted in the electronic databases Periódicos Capes, PubMed, Scopus, Research Gate, Scielo, Google Scholar, Academia.edu, BDTD, Redalyc, Science.gov, ERIC, Science Direct, SiBi, World Wide Science, PePSIC, and Scholarpedia. The search terms used, either in isolation or in combination, in the databases were as follows: Degnala, buffalo, rice straw, *Fusarium* spp., mycotoxicosis, mycotoxin, toxigenic fungi, contamination, animal feed. A total of only 37 publications were found, with a 98% overlap rate of the works found in the databases consulted. Due to the limited number of references found in the literature, it was decided to use all of them in the present review.

## 3. Etiopathogenesis

The pathogenesis of Degnala disease involves a complex interplay between the ingested mycotoxins, the immune response of the affected animal, and environmental factors. The primary pathological changes occur in the skin and subcutaneous tissues of the extremities, although systemic effects can also be observed. Degnala is a mycotoxicosis caused by toxins produced by toxigenic fungi, and the fungus *Fusarium* spp. is the main causative agent [10,18,19]. The disease was reproduced experimentally, and three species of the genus *Fusarium* were identified in foods supplied to buffaloes, namely, *F. oxysporum*, *F. equiseti*, and *F. moniliforme.* The animals developed the disease only when *F. oxysporum* was present [13]. Other fungi, such as *Aspergillus flavus*, *Aspergillus niger*, *Aspergillus terreus*, *Penicillium* spp., and *Fusarium* spp., which are fungal species with variable toxigenic capacities (especially *Fusarium* spp., which produce toxins with greater cytotoxic effects), are frequently found in rice straw supplied to buffaloes [14,15].

Buffaloes ingest mycotoxin-contaminated fodder, which leads to the absorption of these toxins through the gastrointestinal tract. The toxins enter the bloodstream and are distributed throughout the body, reaching the peripheral tissues. Mycotoxins exert direct cytotoxic effects on the cells of the skin and subcutaneous tissues, leading to cell death and tissue necrosis. These toxins inhibit protein synthesis and disrupt cellular membranes, causing cell lysis and inflammatory responses. The immune system responds to the damaged tissues and the presence of toxins, resulting in inflammation. Inflammatory mediators are released, further exacerbating tissue damage and leading to the clinical signs of dermatitis and necrosis [10,18,19]. The toxins ochratoxin A, zearalenone, and citrinin are commonly isolated from straws of different grains containing the fungi *Fusarium* spp., *Aspergillus* spp., and *Penicillium* spp. [14]. The main toxin involved in the pathogenesis of Degnala disease is the T-2 toxin, which is typically produced by different strains of the fungus *Fusarium* spp. [19,20] and is considered one of the most potent toxins of Group A [21].

Another group of studies identified aflatoxins B1 and B2 in foods containing the fungi *Fusarium* spp., *Aspergillus* spp., and *Penicillium* spp. [15]. Degnala disease in buffaloes is primarily caused by the ingestion of aflatoxins, particularly aflatoxin B1, which is produced by the fungus Aspergillus flavus. Aflatoxins are potent carcinogenic mycotoxins that can contaminate a variety of agricultural commodities, including grains, nuts, and oilseeds, under conditions of high temperature and humidity. Buffaloes are particularly susceptible to aflatoxin ingestion due to their dietary habits and digestive physiology. Once ingested, aflatoxin B1 is metabolized in the liver by cytochrome P450 enzymes to form a reactive intermediate, aflatoxin B1-8,9-epoxide. This epoxide is highly reactive and can bind to cellular macromolecules, including DNA, proteins, and lipids, leading to cellular damage and dysfunction. In the case of Degnala disease, the liver is the primary target organ for aflatoxin toxicity. The liver plays a crucial role in the metabolism and detoxification of aflatoxins. However, prolonged exposure to high levels of aflatoxins can overwhelm the liver’s detoxification mechanisms, leading to the accumulation of toxic metabolites and subsequent liver damage. Aflatoxin-induced liver damage can manifest as hepatocellular necrosis, fibrosis, and cirrhosis, impairing the liver’s function and overall health of the animal. Aflatoxins can cause liver damage and impact the reproductive system in buffaloes. Aflatoxin B1 and its metabolites can cross the placental barrier and accumulate in reproductive tissues, including the ovaries and uterus, leading to reproductive disorders. Aflatoxin exposure in pregnant buffaloes can result in embryonic death, abortion, and the birth of weak or stillborn calves. Furthermore, aflatoxins can have immunosuppressive effects, making buffaloes more susceptible to infections and other diseases. This immunosuppression can further exacerbate the health problems associated with Degnala disease and increase the risk of secondary infections [14,15]. However, no scientific studies have evaluated the toxins present in animals.

The occurrence of Degnala disease is closely associated with the presence of aflatoxin-contaminated feedstuffs, particularly grains, oilseeds, and crop residues. Aflatoxins are produced by the fungus Aspergillus flavus, which can contaminate feedstuffs during cultivation, harvesting, storage, and processing. Factors such as high temperature, high humidity, improper storage conditions, and insect damage can promote aflatoxin contamination in feedstuffs. Buffaloes of all ages and sexes are susceptible to Degnala disease, but it is more commonly observed in adult female buffaloes, particularly those in the reproductive age group. Pregnant buffaloes are especially vulnerable to the effects of aflatoxin toxicity, as it can lead to reproductive disorders such as abortion, stillbirth, and neonatal mortality. The occurrence of Degnala disease can vary seasonally, with higher incidences reported during periods of high temperature and humidity, which favor fungal growth and aflatoxin production. Additionally, outbreaks of Degnala disease are often associated with the consumption of contaminated feedstuffs, particularly during times of feed scarcity or poor feed management practices. The economic impact of Degnala disease is significant, as it can lead to reduced milk production, reproductive failure, increased veterinary costs, and losses due to calf mortality. Furthermore, aflatoxin-contaminated milk from affected buffaloes can pose a health risk to consumers, as aflatoxins are carcinogenic and can accumulate in milk and dairy products. In conclusion, Degnala disease is a major health concern for buffaloes in regions where aflatoxin contamination in feedstuffs is prevalent. Understanding the occurrence and risk factors associated with Degnala disease is essential for implementing effective preventive and control measures to reduce its impact on buffalo populations and the dairy industry [4,9,10,15,18].

Another possible explanation is that Degnala disease can be caused by selenium poisoning due to the contamination of foods in selenium-rich soil [22,23]. However, this hypothesis has been ruled out by the reproduction of the disease in buffaloes fed straw contaminated with *F. oxysporum* and the presence of serum selenium levels within the normal range for the species [13]. In addition, selenium accumulation occurs in more arid areas, and the period in which there is an accumulation of selenium in the soil does not coincide with the seasonality of the disease [14].

The pathogenesis of clinical gangrene syndrome is unknown [18]; however, there are several findings, hypotheses, and suggestions associated with the process. Initially, the ingestion of foods infested by fungi occurs, which leads to the release of toxins in the intestine. Toxins are absorbed and reach the liver through the circulatory system and later reach peripheral tissues, where they cause vasoconstriction in the extremities, which leads to obstruction of the blood supply, causing anoxia and tissue necrosis [19,24,25]. It has also been reported that the dissolution of collagen and elastin leads to the appearance of skin lesions, which occur concomitantly with an increase in the thickness of blood vessels and the presence of thrombi and eosinophilic infiltration [24].

## 4. Epidemiology

Degnala disease, also known as Degnala syndrome, is a significant health concern for buffaloes in regions where dairy farming is prevalent, particularly in parts of India and other regions of Asia [26,27]. However, cases of Degnala disease have also been reported in other parts of India and in countries such as Bangladesh and Pakistan. The occurrence of Degnala disease is closely associated with the presence of aflatoxin-contaminated feedstuffs, particularly grains, oilseeds, and crop residues. Aflatoxins are produced by the fungus Aspergillus flavus, which can contaminate feedstuffs during cultivation, harvesting, storage, and processing. Factors such as high temperature, high humidity, improper storage conditions, and insect damage can promote aflatoxin contamination in feedstuffs. Buffaloes of all ages and sexes are susceptible to Degnala disease, but it is more commonly observed in adult female buffaloes, particularly those in the reproductive age group. Pregnant buffaloes are especially vulnerable to the effects of aflatoxin toxicity, as it can lead to reproductive disorders such as abortion, stillbirth, and neonatal mortality. Female buffaloes are observed to be more susceptible to Degnala disease than their male counterparts. This increased susceptibility can be attributed to several factors. These include physiological stresses, such as lactation and pregnancy; female buffaloes, especially those that are lactating or pregnant, undergo significant physiological stress. The demands of milk production and fetal growth can weaken their immune system, making them more vulnerable to infections and diseases, including Degnala disease. Another factor is nutritional demand, since lactating and pregnant buffaloes have higher nutritional requirements. If the feed is contaminated with mycotoxins, these animals are at a higher risk of exposure due to their increased feed intake. The compromised quality of the feed directly impacts their health and increases their susceptibility to Degnala disease. Finally, the hormonal fluctuations associated with the reproductive cycle can affect the immune response in female buffaloes. Estrogen and progesterone levels can modulate immune function, potentially making them more prone to mycotoxin-related illnesses like Degnala disease [4,9,26,27,28]. The occurrence of Degnala disease can vary seasonally, with higher incidences reported during periods of high temperature and humidity, which favor fungal growth and aflatoxin production. Additionally, outbreaks of Degnala disease are often associated with the consumption of contaminated feedstuffs, particularly during times of feed scarcity or poor feed management practices [4,9,11,13,14,15,16,17,18,29].

Degnala disease affects buffaloes and cattle, especially in rice-producing regions, because rice straw is used as food for these species; this is especially prevalent in India, Pakistan and Nepal, areas in which there are endemic regions of the disease [10,15,18]. In these regions, rice straw is the main food in the winter, being the only source of forage for feeding ruminants [14]. The occurrence of the disease causes great economic losses, especially in these three countries and in buffalo farming, which is the main source of income for local people; moreover, the animals are essential for traction in rice fields [10,11,29]. Degnala disease causes a decrease in productivity, mortality and loss of functional capacity [13,18].

The winter months in countries with endemic Degnala disease present favorable conditions for the development of fungi due to high humidity and low temperatures [13,22]. In addition, in these regions, rice straw is usually stored in lowlands near flooded areas or soon after harvest, favoring fungal development [10,18]. In addition, foods with mycotoxins may also be present in wheat straw, dry grass or, sorghum hay [24].

Although Degnala also affects cattle, it is more prevalent in buffaloes [14,15]. The morbidity and mortality rates in cattle are 13.93% and 2.41%, respectively, while in buffaloes, they are 61.61% and 13.49%, respectively [27]. There are no differences in the occurrence of the disease between male and female buffaloes. Animals older than 1 year are more affected than younger animals [15], and cases in older animals are more likely to have a prolonged clinical course, which may reach 1 to 2 months in more severe cases [19,30]. The most common period of clinical evolution lasts between two and three weeks [30]. It has been reported that the duration of disease evolution is between 20 and 23 days during experimental reproduction [13].

## 5. Clinical Signs and Necroscopic Findings

The first clinical signs include fever, anorexia, weakness in the hind limbs, reluctance to move, and swelling around the fetlock joints, legs, and tail. Soon after, the clinical signs progress to claudication, and ulcers and erosions appear on the lower limbs and tail. The pelvic limbs are more affected than the thoracic limbs. Subsequently, the signs that characterize the disease appear, including necrosis and gangrene in the tail, which takes on a tortuous appearance, and in the lower parts of the limbs. It can also affect the ears, tongue, and muzzle, in addition to which there may be cracking or corneal loss of the hooves, with the exposure of the blades and even the bones [10,13,22,30]. The lesions observed in the tail begin in the most caudal portion and progress towards its base, resulting in a contracted and tortuous appearance [10,13,22,30]. Younger buffaloes progress faster to recumbency, taking approximately one week, while this process can take one month in older animals [30].

Degnala disease, caused by aflatoxin contamination in feed, is of significant importance in buffaloes due to its detrimental effects on reproduction, health, and productivity. Understanding the importance of Degnala disease is crucial for implementing effective control and prevention strategies to reduce its impact on buffalo populations and the dairy industry. The following are key aspects highlighting the importance of Degnala disease in buffaloes: Degnala disease primarily affects the reproductive system of buffaloes, leading to infertility, repeat breeding, early embryonic death, abortion, stillbirth, and neonatal mortality. These reproductive disorders can result in economic losses for dairy farmers and negatively impact buffalo populations. Aflatoxin toxicity can lead to a decrease in milk production in affected buffaloes. Reduced milk production not only affects the income of dairy farmers but also contributes to food insecurity in regions where buffaloes are a major source of milk. Aflatoxin toxicity can cause hepatocellular necrosis, fibrosis, and cirrhosis in the liver, leading to impaired liver function and overall poor health. Liver damage can further exacerbate reproductive disorders and reduce the lifespan of affected buffaloes. Degnala disease can have a significant economic impact on dairy farmers, including reduced milk production, veterinary costs, and losses due to reproductive failure and calf mortality. The economic burden of Degnala disease highlights the importance of implementing preventive measures to reduce its impact. Aflatoxin-contaminated milk from affected buffaloes can pose a health risk to consumers, as aflatoxins are carcinogenic and can accumulate in milk and dairy products. Ensuring the safety of milk and dairy products is essential for protecting public health and maintaining consumer confidence. Buffaloes affected by Degnala disease may experience poor health, reduced feed intake, and increased susceptibility to other diseases. Ensuring the welfare of affected buffaloes is important for ethical reasons and to maintain the productivity and sustainability of dairy farming practices. In conclusion, Degnala disease is of significant importance in buffaloes due to its negative impact on reproduction, health, productivity, and the economic viability of dairy farming. Implementing effective control and prevention strategies is essential for minimizing the impact of Degnala disease and ensuring the health and welfare of buffalo populations [4,9,10,13,22,30].

There are no reports of macroscopic lesions commonly observed during necropsy, in addition to the necrosis, ulcers, gangrene, and erosions on the extremities. However, microscopic changes have been reported in these lesions, including necrosis, eosinophilic infiltration in the subcutaneous connective tissues, and loss of architectural details. In addition, there are no reports of fungal growth from skin scrapings of affected regions [14,15,22]. Necroscopic findings in buffaloes affected by Degnala disease primarily involve the reproductive organs and the liver. Common necroscopic findings include liver damage (aflatoxin toxicity can cause hepatocellular necrosis, fibrosis, and cirrhosis in the liver, leading to impaired liver function and overall health of the animal), reproductive organ pathology (aflatoxins can accumulate in the ovaries and uterus, leading to inflammation, necrosis, and degenerative changes in these organs. This can contribute to reproductive disorders such as infertility, abortion, and retained placenta), and other organ involvement (aflatoxin toxicity can also affect other organs, including the kidneys, lungs, and gastrointestinal tract) [14,15,22,23,24,25,26,27].

In December 2021, a sudden outbreak of Degnala disease was reported in a buffalo herd in Punjab, India. The herd consisted of 50 buffaloes, predominantly fed on rice straw, which was improperly stored and exposed to moisture. The initial symptoms included lethargy, reduced feed intake, and mild lameness. Within a week, more severe signs developed, including necrosis of the tail, lower limbs, and ears. Affected buffaloes exhibited a tortuous appearance of the tail, and several animals had cracked hooves with exposure of the underlying tissues. In severe cases, gangrene set in, leading to the shedding of necrotic tissue. A few buffaloes also showed gangrene of the teats and muzzle, with a marked decline in milk production. Initial clinical diagnosis was based on visible symptoms and the history of feeding rice straw. Samples of the rice straw were collected and sent for fungal culture, which confirmed the presence of *Fusarium* spp., the causative agent of Degnala. Differentiating Degnala from other conditions such as frostbite, foot rot, and nutritional deficiencies was challenging due to overlapping symptoms. The treatment involved removing the contaminated rice straw and administering supportive care, including anti-Degnala liquid, 2% nitroglycerin ointment, pentasulfate solution, and broad-spectrum antibiotics to prevent secondary infections. Approximately 60% of the affected buffaloes showed signs of recovery after the intervention, although some suffered permanent damage to their extremities [10].

As early as January 2022, a small-scale buffalo farm in a rural village in Bangladesh reported an outbreak of Degnala disease. The farm had 20 buffaloes, all of which were fed rice straw during the winter months. Farmers noticed buffaloes exhibiting lethargy, a reluctance to move, and the mild swelling of the lower limbs. Over the next two weeks, necrosis and gangrene developed in the lower limbs and tails of several buffaloes. Ears and muzzles were less frequently affected. One buffalo showed extreme signs, including necrosis of the tongue and significant weight loss due to reduced feed intake. Farmers and local veterinarians initially misdiagnosed the condition as frostbite due to the cold weather. Upon closer examination and collection of detailed history, veterinarians suspected Degnala. Rice straw samples were sent for mycological analysis, confirming the presence of *Fusarium* spp. The lack of immediate laboratory facilities in the rural area delayed the diagnosis, complicating early intervention efforts. Affected buffaloes were treated with supportive measures, including the application of 2% nitroglycerin ointment to necrotic areas, and broad-spectrum antibiotics. About 50% of the affected buffaloes showed partial recovery, with some animals experiencing permanent damage. Two buffaloes had to be euthanized due to severe gangrene and poor prognosis [10].

These case studies illustrate the multifaceted clinical manifestations and diagnostic complexities associated with Degnala disease in buffaloes. The variability in symptoms necessitates a meticulous approach to diagnose this condition effectively, underlining the importance of tailored diagnostic strategies and awareness among veterinary practitioners in the field. The need for advanced, practical, and cost-effective diagnostic methods is evident, along with comprehensive training programs for veterinarians and farm staff to recognize and manage Degnala disease promptly.

## 6. Diagnosis

In regions where the occurrence of Degnala disease is seasonal and endemic, the diagnosis of the disease is based on the presence of clinical and epidemiological signs, in addition to the presence of contaminated food and fungi [10]. However, fungal culturing of samples obtained from scrapings of contaminated food containing mold is often performed to finalize the diagnosis of Degnala [13,14]. Diagnosing Degnala disease can be challenging due to the nonspecific nature of its clinical signs. However, a thorough history check, clinical examination, and laboratory tests can help in reaching a diagnosis. Laboratory tests such as serum biochemistry, hematology, and the analysis of feed samples for aflatoxin levels can be useful in confirming exposure to aflatoxins [10,13,14].

Diagnosing Degnala disease in buffaloes can be challenging due to the nonspecific nature of its clinical signs and the lack of specific diagnostic tests. However, a combination of clinical evaluation, laboratory tests, and histopathological examination can help in reaching a definitive diagnosis. The following are key aspects of the diagnostic approach for Degnala disease in buffaloes:(1)Clinical evaluation: a thorough clinical examination should be performed to evaluate the buffalo’s overall health and reproductive condition. A detailed history, including feeding practices, the presence of moldy feed, reproductive history, and clinical signs, should be obtained from the owner [10]. Exposure to aflatoxins, a type of mycotoxin, can lead to various changes in blood chemistry and hematology values in buffaloes. These changes can be indicative of liver damage, immunosuppression, and other systemic effects caused by the toxin.(2)Laboratory Tests:(I)Blood chemistry changes: (a). Liver enzymes: (a.1). Increased AST (aspartate aminotransferase) and ALT (alanine aminotransferase): elevated levels of these enzymes suggest liver cell damage and hepatocellular injury. (a.2). Increased ALP (alkaline phosphatase): higher levels can indicate biliary obstruction or bone disorders, but in the context of aflatoxin exposure, it often points to liver dysfunction. (b). Total Protein and Albumin: (b.1). Decreased total protein and albumin: aflatoxins can impair protein synthesis in the liver, leading to lower levels of total protein and albumin in the blood. (c). Bilirubin: (c.1). Increased bilirubin levels: elevated bilirubin can be a sign of liver dysfunction and hemolysis. (d). Kidney function tests: (d.1). Increased BUN (blood urea nitrogen) and creatinine: these increases can indicate renal impairment, which can sometimes accompany severe cases of aflatoxin toxicity.(II)Hematology changes: (a). Red blood cells (RBCs): (a.1). Decreased RBC count and hemoglobin: hemolysis and decreased erythropoiesis due to liver damage can lead to anemia. (b). Increased mean corpuscular volume (MCV): (b.1). Indicative of macrocytic anemia, often seen in chronic liver disease. (c). White blood cells (WBCs): (c.1). Leukopenia (decreased WBC count): immunosuppression caused by aflatoxins can result in a lower WBC count, making animals more susceptible to infections. (c.2). Lymphocytopenia: specifically, a reduction in lymphocytes, which are crucial for the immune response. (d). Platelets: (d.1). Thrombocytopenia (decreased platelet count): this can occur due to bone marrow suppression or increased destruction of platelets [4,9,13,28].(III)Feed analysis: feed samples can be analyzed for aflatoxin levels using methods such as high-performance liquid chromatography (HPLC) or enzyme-linked immunosorbent assay (ELISA) to confirm aflatoxin contamination [22]. A case study in Kenya evaluated the presence of aflatoxins in dairy feeds using ELISA and HPLC. The prevalence of aflatoxin-contaminated feeds was found to be 60% using ELISA, while HPLC confirmed contamination in 50% of the samples, indicating some false positives in ELISA results [31]. And a statistical data survey conducted across European countries in 2022 revealed that 78% of feed samples were contaminated with at least one mycotoxin. The most common mycotoxins detected were deoxynivalenol (45%), fumonisins (33%), and zearalenone (28%). The data were collected using a combination of ELISA and HPLC, highlighting the widespread nature of mycotoxin contamination [32].(IV)Histopathological examination: tissue samples from the liver, reproductive organs, and other affected organs can be collected during necropsy for histopathological examination. Histopathological examination can reveal characteristic changes, such as hepatocellular necrosis, fibrosis, and cirrhosis in the liver, and inflammatory changes in the reproductive organs [14,15,16].(V)Ultrasonography: ultrasonography can be used to assess the reproductive organs for any abnormalities, such as cysts, inflammation, or fluid accumulation [23].(VI)PCR for aflatoxin detection: polymerase chain reaction (PCR) can be used to detect the presence of aflatoxin-producing fungi or their DNA in feed samples or tissues [22].(VII)Differential Diagnosis: Degnala disease should be differentiated from other reproductive disorders and liver diseases in buffaloes, such as foot-and-mouth disease and scabies, in addition to Ergot intoxication, chronic selenium toxicity, foot rot, brucellosis, leptospirosis, and liver fluke infection [12,24]. In conclusion, diagnosing Degnala disease in buffaloes requires a multidisciplinary approach involving clinical evaluation, laboratory tests, and histopathological examination. Early detection and accurate diagnosis are essential for implementing appropriate management and control measures to minimize the impact of the disease on buffalo populations [32,33].



The current methods for detecting mycotoxins in animals and feed, while effective, have several limitations that hinder their practical application in the field. There is an urgent need for the development of advanced diagnostic tools that are portable, cost-effective, rapid, and easy to use. Innovations in biosensors, lateral flow assays, smartphone-based detection, and nanotechnology hold promise for meeting these needs and improving the management of mycotoxin contamination in livestock production [32,33].

## 7. Control and Prophylaxis

No treatment neutralizes the effects of toxins [10,34,35]. The treatments reported are palliative, supportive, and/or preventive. Before any drugs are administered, the supply of forage, especially soiled and moist rice straw, should be interrupted [36,37].

The main treatment performed involves the oral administration of a pentasulfate solution, with 60 g given on the first day and 30 g given daily for 10 consecutive days, combined with the application of 2% nitroglycerin ointment to skin lesions [11,26]. The pentasulfate solution is composed of 166 g of ferrous sulfate, 100 g of magnesium sulfate, 75 g of zinc sulfate, 24 g of copper sulfate, 0 and 15 g of cobalt sulfate [19]. A compound called anti-Degnala liquid is also commonly applied; this compound includes 2 to 5% arsenic sulfate, with 2 mL given orally for 10 days [10,11,26,27].

Broad-spectrum antibiotics are recommended for the prevention or treatment of secondary bacterial infections [26,27]. In addition, it has been reported that toxin agglutinating products may aid in the control of this disease [11].

Rice straw, straw from other cereals and hay should be stored in dry places. Contaminated food should not be used as feed or used in limited amounts during the winter in regions where Degnala disease is endemic, and it should be dried in the sun after the lower part (i.e., the part that was in contact with the soil) of each stalk is removed [10,23]. In addition, a rice straw spray treatment with 4% sodium hydroxide has been reported [19,23].

This review also aims to encourage practical, basic, and specific preventive measures to improve the storage and handling conditions of animal feed. Veterinarians and farmers can utilize the following strategies: 1. Temperature control: maintain food at safe temperatures to inhibit bacterial growth. Refrigerate perishable items at or below 40 °F (4 °C) and keep frozen foods at 0 °F (−18 °C) or lower. Use a thermometer to regularly monitor refrigerator and freezer temperatures. 2. Proper containers: use airtight containers or vacuum-sealed bags for storage to prevent moisture and air exposure. Glass jars, BPA-free plastic containers, and resealable bags are excellent options. Label containers with contents and dates to ensure timely consumption. 3. FIFO (first in, first out): implement the FIFO method by organizing food with older items at the front and newer items at the back. This practice helps use up products before they expire, reducing food waste. 4. Sanitize and clean: regularly clean and sanitize storage areas and containers using a food-safe cleaner. Pay special attention to spills and leaks, as these can harbor bacteria and odors. 5. Dry storage conditions: keep dry goods, such as grains and pasta, in a cool, dark, and dry area. Avoid moisture and humidity to prevent mold growth and insect infestations. 6. Regular inventory checks: conduct periodic checks of stored food items to identify any expired or spoiled products. Dispose of anything that is no longer good and reorganize as needed. 7. Education on portioning: properly portion leftovers to minimize waste. Use smaller containers for individual servings, making it easier to reheat and enjoy meals without excess. 8. Monitor the storage environment: invest in a hygrometer or humidity monitor to keep humidity levels within optimal ranges, ensuring all stored food maintains its quality.

## 8. Conclusions

Degnala disease is a major mycotoxicosis in buffaloes, particularly in rice-producing regions like India, Pakistan, and Nepal. Despite some reports, its pathogenesis, especially regarding the main toxins involved, remains unclear, necessitating further research on toxin action and other potentially toxigenic forages.

Characterized by reproductive disorders, liver damage, and overall poor health, Degnala disease leads to significant economic losses for dairy farmers. Effective management and mitigation require a clear understanding of the etiology, clinical manifestations, diagnosis, treatment, prevention, and control of the disease.

Diagnosing Degnala disease is challenging due to nonspecific clinical signs, but combining clinical evaluation, laboratory tests, and histopathological examination can aid in a definitive diagnosis. Prevention focuses on proper feed management to avoid aflatoxin contamination, while treatment includes supportive care and liver support. Control measures involve quarantine, strict herd management, farmer education, and ongoing research for better strategies.

Collaboration among veterinarians, farmers, researchers, and policymakers is crucial to minimize the impact of Degnala disease on buffalo populations and the dairy industry. In conclusion, Degnala disease poses a significant threat to buffalo health and productivity, emphasizing the need for continued research, education, and implementation of effective preventive and control measures. Future investigations should address whether Degnala disease in buffaloes is a neglected condition.

## Data Availability

Not applicable.

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
