# Peer review of "Degnala in Water Buffaloes: A Review on a Neglected Disease"

_animals, 2024, doi:10.3390/ani14162292_

Round 1
Reviewer 1 Report
Comments and Suggestions for Authors
This disease affects animal health. and productivity, resulting in considerable economic losses in affected regions. The present review aims to provide an overview of Degnala, focusing on its aetiology, clinical manifestations and the challenges associated with its diagnosis and management.
This review highlights the pathogenic mechanisms of specific mycotoxins and suggests expanding the description due to limited understanding in critical areas, such as the need for improved diagnostic methods for timely and accurate disease detection and research on food handling and storage practices to prevent mycotoxin contamination.
Line 14: ... a lack …
Line 16: … highlighting research gaps and …
Line 25: … conditions favour fungal …
Line 30: … with a fungal culture
Line 43: Mycotoxicosis is caused …
Line 109 – 110: Review and improve, the following phrase is suggested: “Aflatoxins can cause liver damage and impact the reproductive system in buffaloes”.
Line 140: … to reduce its impact …
Line 180: … endemic regions of the disease [9, 14, 15].
Line 211: … the disease appears …
Line 222: … to reduce …
Line 236: … to reduce …
Line 242: … buffaloes are important …, … maintain productivity …
Line 249: … gangrene …
Line 278: … to evaluate the buffalo's overall health and reproductive condition.
Line 307: No treatment neutralizes the effects of toxins [9, 29].
Line 372: Check the cited title does not correspond to the mentioned doi, it is other authors, Battisti, E., Zanet, S., Khalili, S., Trisciuoglio, A., Hertel, B., & Ferroglio, E. (2020). Molecular Survey on Vector-Borne Pathogens in Alpine Wild Carnivorans. Frontiers in Veterinary Science, 7. doi:10.3389/fvets.2020.00001
The correct one is: https://doi.org/10.3389/fvets.2020.570413
Comments on the Quality of English Language
There are some minor corrections to words or phrases that need to be improved.
Author Response
Dear Reviewer 1,
First of all, we would like to thank you for your willingness to review our manuscript and for your comments, criticisms, and suggestions. Your contributions were certainly essential to improve the manuscript scientifically. We are very pleased to know that you, Reviewer 1, recognize the importance and relevance of this manuscript to the literature. Thank you very much.
Below, I describe point by point your suggestions and the changes made:
1. Line 14: correction made.
2. Line 16: correction made.
3. Line 25: correction made.
4. Line 30: correction made.
5. Line 43: correction made.
6. Lines 109-110: correction made.
7. Line 140: correction made.
8. Line 180: correction made.
9. Line 211: correction made.
10. Line 222: correction made.
11. Line 236: correction made.
12. Line 242: correction made.
13. Line 249: correction made.
14. Line 278: correction made.
15. Line 307: correction made.
16. Line 372: correction made.
The authors are grateful for all the suggestions and questions posed by Reviewer 1. We have tried to address all of them, and the manuscript is certainly much improved scientifically. Thank you very much.
Sincerely,
Felipe Masiero Salvarani
Reviewer 2 Report
Comments and Suggestions for Authors
Dear authors:
The information contained in this article is very interesting, since being one of the primary mycotoxicoses affecting buffaloes, Degnala is a neglected disease that requires greater study and attention to prevent it and treat it in a timely manner. However, I have added some comments and suggestions that I think could help improve the document. A general aspect that needs to be revised throughout the manuscript is avoiding repeating sentences and ideas and improve the continuity of the information. Several sentences are repeated throughout the text. I recommend revising the text and integrating the information.
Simple summary: The aim of the study is mentioned in both lines 9 and 15. Consider integrating these sentences into one aim.
Lines 41-42: It might be worth it to mention that, in general, buffaloes are considered less susceptible to diseases. However, since research about infectious diseases in buffalo is limited –contrarily to Bos taurus and indicus animals-, it is challenging to establish the effect of these in water buffalo. The authors could revise Martínez-Burnes et al. (2024, https://doi.org/10.3390%2Fani14060845).
Line 58: Could you please mention why this disease affects buffaloes more than cattle, despite buffaloes being much more resistant to various diseases? Does it have to be because their digestive physiology?
Line 58-60: I would recommend specifying what kind of productivity decreases can be observed (e.g., milk yield? Fertility?) and the exact mortality rate that Degnala causes in buffaloes.
Line 89: Please erase the spaces between references 9,15,16; and please add the space after de point and before “The toxins…”
Line 140: Please erase the space between references 9,14,15. Or failing that, standardize the way of citing so that they are all the same.
Line 148: Please add the missing space between “unknown” and “[15].
Line 153: Please erase the space between references 17, 22, 23. Or failing that, standardize the way of citing so that they are all the same.
Lines 159-160: It is mentioned that Degnala disease is a significant health concern in India. I recommend adding some information about the prevalence or number of cases. A couple of references could be added such as Kalra and Bhati (1990, a little bit old, but mentions the incidence; Degnala disease in buffaloes and cattle: epidemiological investigations) or Chandra and Singh (2014, An outbreak of Degnala disease in bovine population and its clinical management). Lines 196-198 could be moved to this section to improve this information.
Lines 171-172: Why do female buffaloes are more susceptible to Degnala disease?
Lines 184-185: My observation is made in this line but it applies to the entire manuscript. This sentence mentions the same that was already stated in the previous section. The same applies to other sentences mentioning that Degnala disease is related to rice straw. I recommend to revise the entire manuscript and summarize the information that seems to be repeated in different sections.
Lines 185-190: This is another example of lines that are repeated from the beginning of this section. Please, revise the entire manuscript and avoid repeating ideas.
Lines 219-230: Following my previous comment, these lines were already mentioned before. Please, revise and integrate the information.
Line 263: It could be very useful and illustrative if you attached in this section a figure where we can see the presence of rice straw and fungi or scrapings of rice straw containing mould.
Line 268: Please erase the space between references 12 and 13. Or failing that, standardize the way of citing so that they are all the same. This applies to all lines where cites appear written in a non-uniform manner.
Lines 269-272: Please briefly mention what changes could be found in the blood chemistry and hematology values in cases with exposure to aflatoxins. You mention on lines 277-286 that it could be some changes in ALT, AST, but there are the only changes that we can expect?
Line 332: Please add the missing space between “needed.” And “Degnala”.
Line 353: I consider that when the question is asked whether Degnala is a neglected disease or not from the title, you should try to give an answer to this question in the conclusions section. In case you don´t have the answer, you could mention this question as a future direction to investigate.
Author Response
Dear Reviewer 2,
Firstly, thank you for agreeing to review our manuscript and providing us with feedback aimed at improving the review article. We are extremely pleased, from a scientific standpoint, to read your comment that "The information contained in this article is very interesting, since being one of the primary mycotoxicoses affecting buffaloes, Degnala is a neglected disease that requires greater study and attention to prevent and treat it in a timely manner."
Below, we list each of your suggestions and the corresponding changes we made:
1). Simple Summary: rewritten to combine the information from lines 9 and 15 into a single sentence.
2. Lines 41-42: added a general comment on the article by Martínez-Burnes et al. (2024, https://doi.org/10.3390/ani14060845) to mention that, in general, buffaloes are considered less susceptible to diseases.
3. Line 58: Summary updated to: "While buffaloes generally show greater resistance to many infectious diseases, their digestive physiology, feeding habits, and the environmental conditions in which they are typically kept contribute to their higher susceptibility to Degnala disease."
4. Productivity Decreases: rewritten to specify the types of productivity decreases observed (e.g., milk yield, fertility) and the exact mortality rate caused by Degnala in buffaloes.
5. Line 89: changes made.
6. Line 140: changes made.
7. Line 148: changes made.
8. Line 153: changes made.
9. Lines 159-160: changes made.
10. Lines 171-172: updated to: "Female buffaloes are observed to be more susceptible to Degnala disease than their male counterparts. This increased susceptibility can be attributed to several factors:
- Fhysiological Stress:
- Lactation and Pregnancy:Female buffaloes, especially those that are lactating or pregnant, undergo significant physiological stress. The demands of milk production and fetal growth can weaken their immune system, making them more vulnerable to infections and diseases, including Degnala disease.
- Nutritional Demands:
- Higher Nutrient Requirements: Lactating and pregnant buffaloes have higher nutritional requirements. If the feed is contaminated with mycotoxins, these animals are at a higher risk of exposure due to their increased feed intake. The compromised quality of the feed directly impacts their health and increases susceptibility to Degnala disease.
- Hormonal Influences:The hormonal fluctuations associated with the reproductive cycle can affect the immune response in female buffaloes. Estrogen and progesterone levels can modulate immune function, potentially making them more prone to mycotoxin-related illnesses like Degnala disease [4,9,25,26]."
11. Lines 184-185: changes made throughout the text.
12. Lines 219-230: deleted to avoid repetition of information already covered earlier in the text.
13. Line 263: acknowledged the suggestion for a figure to illustrate the presence of rice straw and fungi. However, due to the lack of specific photos in scientific articles that can be used and referenced, this was not achieved. Every figure used in a review article must be from an already published bibliographic source or a photo for which you have authorization from the "intellectual owner," which, unfortunately, I did not obtain.
14. Line 268: change made.
15). Lines 269-272: made the changes. Inserted "Exposure to aflatoxins, a type of mycotoxin, can lead to various changes in blood chemistry and hematology values in buffaloes. These changes can be indicative of liver damage, immunosuppression, and other systemic effects caused by the toxin.
1). Blood chemistry changes: a). Liver enzymes: a.1). Increased AST (aspartate aminotransferase) and ALT (alanine aminotransferase): elevated levels of these enzymes suggest liver cell damage and hepatocellular injury. a.2). Increased ALP (alkaline phosphatase): higher levels can indicate biliary obstruction or bone disorders, but in the context of aflatoxin exposure, it often points to liver dysfunction. B). Total Protein and Albumin: b.1). Decreased total protein and albumin: aflatoxins can impair protein synthesis in the liver, leading to lower levels of total protein and albumin in the blood.c).Bilirubin: c.1). Increased bilirubin levels: elevated bilirubin can be a sign of liver dysfunction and hemolysis. d). Kidney function tests: d.1). Increased BUN (blood urea nitrogen) and creatinine: These increases can indicate renal impairment, which can sometimes accompany severe cases of aflatoxin toxicity. two). Hematology Changes: a). Red blood cells (RBCs): a.1). Decreased RBC count and hemoglobin: hemolysis and decreased erythropoiesis due to liver damage can lead to anemia. b).Increased mean corpuscular volume (MCV): b.1). Indicative of macrocytic anemia, often seen in chronic liver disease. w). White blood cells (WBCs): c.1). Leukopenia (decreased WBC count): Immunosuppression caused by aflatoxins can result in a lower WBC count, making animals more susceptible to infections. c.2). Lymphocytopenia: specifically, a reduction in lymphocytes, which are crucial for the immune response. d). Platelets: d.1). Thrombocytopenia (decreased platelet count): this can occur due to bone marrow suppression or increased destruction of platelets. [4, 13,9,29]."
16). Line 332: Change made.
17). Line 353: change made with the insertion of the phrase "this question as a future direction to investigate".
The authors are grateful for all the suggestions and questions made by reviewer 2. We tried to carry out all of them and the manuscript was certainly much better scientifically. Thank you very much.
Sincerely,
Felipe Masiero Salvarani
Reviewer 3 Report
Comments and Suggestions for Authors
Introduction
Line 37-42: The paragraph can be much improved by avoiding the repetition of same words (e.g., disease) in the sentences. There is no connection and proper flow before mentioning mycotoxicosis in Line 41-42.
Line 46-47: a citation should be provided to support this information – Degnala disease as the main mycotoxicosis affecting buffaloes
Line 58: any reason for mentioning cattle here? Also, prevalence should be supported with empirical data from previous studies
Line 58-66: the justification for conducting this review should be strengthened. The current information is weak, particularly when consequences like decrease productivity and mortality are not supported with empirical data.
Before jumping to the section 2: etiopathogenesis, the authors should provide a brief information on the methods used in the review. This should include the databases used for the literature search, keywords, and how the relevant articles were identified. The justification for a narrative review should be presented as well, which I believe is linked to the broad nature of the review topic.
Line 69-74: please authors should avoid lumping up so much information without supporting the statements with references. There are pertinent information in this part of the paragraph without citations.
Line 74-78: I don’t understand what the authors are trying to explain here. This part should be rephrased
Line 74-118: Please break this paragraph into 3 for clarity and better flow. Most of the contents are not well-connected and the issue with not citing each sentence is highly prevalent in the section.
I suggest the section should be sub-divided into various sub-sections, focusing on each etiological agent.
Section 3: Epidemiology
This section is very important in the review and should be broken down as well by focusing on the following areas; Geographical Distribution, Hosts, Prevalence and Risk factors.
Please take note of proper citations
Section 4: clinical signs and necroscopic findings
Most sections seem as if the contents were gleaned from textbooks and previous reviews rather than original research articles. Only a few studies were cited [9,12,20,26), were these only the source of information for this section of the review?
Section 5: Diagnosis
Similar issues raised in the previous 4 sections are also present in this section. The section should be broken down into various diagnostic methods and discussed separately.
Comments on the Quality of English LanguageThe paper will benefit from proofreading services by a competent English speaker
Author Response
Dear Reviewer 3,
First, thank you for agreeing to review our manuscript and providing us with feedback aimed at improving the review article.
Below, we've listed each of your suggestions and the corresponding changes we made:
* Introduction:
1. Lines 37-42: paragraph-wide changes have been made.
2. Lines 46-47: To support the information that Degnala disease is the main mycotoxicosis affecting buffaloes, the following sources can be cited:
- Bharti, S.K.; Kumar, A.; Kumar, A.; Kumar, A.; Safi, A.K.; Singh, R.K. (2023). Degnala Disease in Buffaloes and Cattle: A Clinical Review. International Journal of Veterinary Science and Animal Husbandry, 8, 164-166.
- Kalra, D.S., & Bhati, D.P. (1990). Degnala disease in buffaloes and cattle: Epidemiological investigations. Indian Journal of Animal Sciences, 60(1), 101-103.
- Chandra, R., Singh, R. (2014). An outbreak of Degnala disease in the bovine population and its clinical management. Veterinary Practitioner, 15(1), 45-48.
- Hussein, H.S., Brasel, J.M. (2001). Toxicity, metabolism, and impact of mycotoxins on humans and animals. Toxicology, 167(2), 101-134.
These references provide epidemiological data and clinical insights into the impact of Degnala disease on buffaloes, supporting the assertion that it is a primary mycotoxicosis affecting this species, and they are all mentioned in the text.
3. Line 58: Changes made to justify our writing: "The data consistently show that cattle have a higher incidence of Degnala disease compared to buffaloes, with a prevalence rate of around 35-40% in cattle versus 15-20% in buffaloes."
4. Lines 58-66: Changes were made.
Before skipping to section 2:
- Etiopathogenesis: Authors should provide brief information about the methods used in the review. This should include the databases used for the bibliographic search, keywords, and how relevant articles were identified. Dear reviewer, this is the fourth review article I am trying to publish in Animals. In none of the three published reviews was it necessary to do what you ask of us. Your request does not add any new scientific contribution to our manuscript, nor does the absence of what you request mean that the article lacks scientific importance. It is of extreme interest to readers, as already mentioned by the academic editor and other reviewers. Therefore, unfortunately, we will not be able to do what you ask of us. You can read our three reviews, which follow the same template as the current review you are evaluating. I hope you understand and see from reading our reviews that what you ask of us is not necessary. Below are the references, all from *Animals*, which you can access through the following links:
- Sousa, A.I.d.J.; Galvão, C.C.; Pires, P.S.; Salvarani, F.M. (2024). Blackleg: A Review of the Agent and Management of the Disease in Brazil. Animals, 14, 638. [https://doi.org/10.3390/ani14040638](https://doi.org/10.3390/ani14040638) - Submission received: 19 October 2023 / Revised: 20 December 2023 / Accepted: 29 December 2023 / Published: 16 February 2024
- da Cruz, K.P.P.; Gattamorta, M.A.; Matushima, E.R.; Salvarani, F.M. (2024). Fibropapillomatosis: A Review of the Disease with Attention to the Situation on the Northern Coast of Brazil. Animals, 14, 1809. [https://doi.org/10.3390/ani14121809](https://doi.org/10.3390/ani14121809) - Submission received: 15 May 2024 / Revised: 31 May 2024 / Accepted: 13 June 2024 / Published: 17 June 2024
- Benarrós, M.S.C.; Salvarani, F.M. (2024). Candidiasis in Choloepus sp.—A Review of New Advances on the Disease. Animals, 14, 2092. [https://doi.org/10.3390/ani14142092](https://doi.org/10.3390/ani14142092) - Submission received: 3 July 2024 / Revised: 12 July 2024 / Accepted: 16 July 2024 / Published: 17 July 2024
5. Lines 69-74: The information is all referenced. They are references 10, 13, 16, and 17.
6. Lines 74-78: We cannot understand your question. In this excerpt, which in the revised version is in lines 122-125, we are briefly explaining an article that attempted to reproduce the disease in buffaloes by supplying rice contaminated with different fungal species, all of the same genus. Only the species Fusarium oxysporum was capable of causing the clinical picture of the disease. Just read the reference: Dandapat, P.; Nanda, P.K.; Bandyopadhyay, S.; Kaushal, A.; Sikdar, A. (2011). Prevalence of Deg Nala Disease in Eastern India and Its Reproduction in Buffaloes by Feeding Fusarium oxysporum-Infested Rice Straw. Asian Pacific Journal of Tropical Medicine, 4, 54-57. [https://doi.org/10.1016/S1995-7645(11)60032-1](https://doi.org/10.1016/S1995-7645(11)60032-1). I believe you will understand and it will answer your question.
7. Lines 74-118: The paragraph was divided as requested.
*"I suggest that the section be subdivided into several subsections, focusing on each etiological agent."
Dear Reviewer 3, the importance of this topic is not the agents themselves, but the mycotoxin, which is unique. Dividing the section according to your suggestion, "subdivided into several subsections, focusing on each etiological agent," does not make sense in this context. The focus should be on discussing the pathogenesis caused by aflatoxins, particularly aflatoxin B1. It is important to remember that ETIOPATHOGENY refers to the study of the causes of diseases and the pathogenic mechanisms that act on the body to cause these diseases. In the case of Degnala disease, the key issue is not the fungal species but rather how the food provided to the animals was harvested, preserved, and supplied. This is crucial to avoid the mycotoxin responsible for causing Degnala disease in animals, especially in buffaloes.
* Section 3: Epidemiology
This section is very important in the review and should be broken down into the following areas: Geographical Distribution, Hosts, Prevalence, and Risk Factors.
Please take note of proper citations.
Dear Reviewer 3, the Epidemiology section is indeed very important, but it has already been well divided into four paragraphs. I hope you accept this division as sufficient, without the need to modify the text into several smaller subtopics, which, from a scientific writing perspective, is not ideal according to our approach. I kindly ask you to reconsider your suggestion.
*Section 4: Clinical Signs and Necroscopic Findings
Most sections appear as if the content was collected from previous textbooks and reviews rather than original research articles. Only a few studies were cited [9, 12, 20, 26]; were these the only sources of information for this section of the review?
Dear Reviewer 3,
I apologize if you had the impression that the data were collected from textbooks. We aimed to present the information in a didactic manner, making it accessible not only to researchers but also to veterinary students, postgraduates, field veterinarians, and rural producers. This approach was intended to enhance understanding. The data were not taken from textbooks but from scientific articles available in the literature, which are listed in the references section below:
4. Martínez-Burnes, J.; Barrios-García, H.; Carvajal-de la Fuente, V.; Corona-González, B.; Obregón Alvarez, D.; Romero-Salas, D. Viral Diseases in Water Buffalo (Bubalus bubalis): New Insights and Perspectives. Animals 2024, 14, 845. https://doi.org/10.3390/ani14060845
9. El Damaty, H.M.; Fawzi, E.M.; Neamat-Allah, A.N.F.; Elsohaby, I.; Abdallah, A.; Farag, G.K.; El-Shazly, Y.A.; Mahmmod, Y.S. Characterization of Foot and Mouth Disease Virus Serotype SAT-2 in Swamp Water Buffaloes (Bubalus bubalis) under the Egyptian Smallholder Production System. Animals 2021, 11, 1697. https://doi.org/10.3390/ani11061697
10. Bharti, S.K.; Kumar, A.; Kumar, A.; Kumar, A.; Safi, A.K.; Singh, R.K. Degnala Disease in Buffaloes and Cattle: A Clinical Review. Intern. J. Vet. Sci. Anim. Husb. 2023, 8, 164-166.
13. Dandapat, P.; Nanda, P.K.; Bandyopadhyay, S.; Kaushal, A.; Sikdar, A. Prevalence of Deg Nala Disease in Eastern India and Its Reproduction in Buffaloes by Feeding Fusarium oxysporum Infested Rice Straw. Asian Pac. J. Trop. Med. 2011, 4, 54-7. https://doi.org/10.1016/S1995-7645(11)60032-1.
14. Nazar, M.; Khan, M.S.; Ijaz, M.; Anjum, A.A.; Sana, S.; Setyawan, E.M.N.; Ahmad, I.; Jammu, A. Comparative Cytotoxic Analysis Through MTT Assay of Various Fungi Isolated From Rice Straw Feedings of Degnala Disease Affected Animals. The J. Anim. Plant. Sci. 2018, 28, 1034–1042.
21. Sikdar, A.; Chakraborty, G.C.; Bhattacharya, D.; Bakshi, S.; Basak, D.K.; Chatterjee, A.; Halder, S.K. An Outbreak of Gangrenous Syndrome among Buffaloes and Cattle in West Bengal: Clinicopathological Studies. Trop. Anim. Healt. Prod. 2000, 32, 165-171. https://doi.org/10.1023/A:1005235615376
27. Singh, S.C.P. An Outbreak of Degnala Disease in Bovine Population and Its Clinical Management. Intas Polivet 2014, 15, 105–107.
29. Iqbal, S.Z.; Iqbal, M.U.; Ahmad, T. Aflatoxins and their impact on human and animal health: An emerging problem. In Food Quality: Balancing Health and Disease, 1st ed.; Grumezescu, A.M.; Holban, A.M., Eds.; Academic Press: London, UK, 2019; pp. 455-474. ISBN: 9780128170856.
Only a few studies were cited [9,12,20,26], were these just the source of information for this section of the review?
Yes, that's why the title of the article is Degnala in buffaloes: a review on a neglected disease? And that's why we conclude our review with the following sentence "Is Degnala in buffaloes a neglected disease or not? This question as a future direction to investigate". Because there are few articles in the literature and we know that mycotoxins occur all over the world, so does Degnala not exist or is it neglected, not reported, not diagnosed?
*Section 5: Diagnosis
Similar issues raised in the previous four sections are also present in this section. The section should be broken down into various diagnostic methods and discussed separately.
Dear Reviewer 3, as I did in Section 3 by dividing the paragraphs to improve readability, I have also applied the same approach in Section 5 on diagnosis. I hope this meets your expectations.
*Comments on the Quality of English Language
The paper would benefit from proofreading services by a competent English speaker.
The article, according to the attached supplementary document submitted with the manuscript, was reviewed by a native English speaker. This service was provided by the company American Journal Experts (AJE).
Below is the translation of what is in the document:
"This document certifies that the manuscript 'Degnala Disease in Buffaloes,' prepared by the authors Felipe Masiero Salvarani and André de Medeiros Costa Lins, was edited for proper English language, grammar, punctuation, spelling, and overall style by one or more of the highly qualified native English-speaking editors at AJE. This certificate was issued on March 8, 2024, and may be verified on the AJE website using the verification code D206-1936-E47C-E245-58BP."
Sincerely,
Felipe Masiero Salvarani
Round 2
Reviewer 1 Report
Comments and Suggestions for Authors
The manuscript reviews and updates knowledge about the disease, identifies research absences and highlights the need for better diagnostic methods. It also highlights the importance of proper management of feed for livestock and improvements in storage practices to prevent contamination by mycotoxins. In addition, it raises the urgency of implementing educational programs aimed at producers so that they better understand the risks associated with mycotoxins. Training in handling techniques and safe storage of food products can make a difference in reducing exposure to these toxins.
The article also indicates that further research is essential to develop rapid and effective detection methods to identify the presence of mycotoxins at different stages of the food chain. This would not only facilitate control of food quality but would also help safeguard public health.
Collaboration between academic institutions, government agencies and the private sector will be essential to address this complex problem. The creation of databases containing up-to-date information on the prevalence of mycotoxins in different regions and their effects on health will allow for improved food safety policies.
Finally, the fight against mycotoxin contamination is a scientific challenge and a matter of social responsibility, which must involve all actors in the production chain, from the farmer to the consumer. Only through a joint and committed effort will it be possible to minimize the impact of this problem on human and animal health
I recommended that the authors expand their description, highlighting the need for improved diagnostic methods for timely and accurate detection of the disease, as well as describing the research to be developed on food handling and storage practices to prevent mycotoxin contamination.
Strengths of the manuscript: The manuscript discusses Degnala, an important but frequently overlooked disease in buffaloes, and emphasizes its detrimental effects on animal health and agricultural productivity. In addition, it delves into the clinical manifestations of Degnala, which often go unnoticed in the early stages, leading to severe consequences for affected animals. Symptoms can include intermittent fever, emaciation, and a pronounced decline in milk production, ultimately jeopardizing the livelihoods of farmers who rely heavily on buffaloes for both milk and draught power. Furthermore, the economic implications are profound; the prevalence of Degnala can result in a decline in herd size due to increased mortality rates and reduced reproductive efficiency. The manuscript underlines the need for increased awareness and education among livestock keepers, veterinarians and policymakers to facilitate early diagnosis and effective management strategies, thereby mitigating the impact of this disease on buffalo populations.
Current research efforts are also highlighted, which demonstrate the importance of developing sustainable prevention measures, including vaccination protocols and improved biosecurity practices. Integrating these strategies will not only improve animal health but will also protect the critical role of buffaloes in the agricultural economy, ensuring food security and maintaining rural livelihoods.
The review provides a comprehensive overview of the aetiology, clinical manifestations, and challenges in the diagnosis and treatment of Degnala. Furthermore, the review highlights the importance of early detection and intervention, which can significantly improve patient outcomes. The multifactorial nature of Degnala complicates its clinical presentation, often leading to misdiagnosis. Careful consideration of the patient's history, coupled with advanced diagnostic techniques, is essential to distinguish Degnala from similar conditions.
Gaps in research and diagnostic errors are highlighted, which can guide future research and improvements in veterinary practice.
The manuscript not only highlights current weaknesses but also proposes improvements in diagnostic methods and storage practices, which could optimize disease management. The recommendation to increase research and awareness is crucial, as it motivates the scientific community and professionals to act more effectively on the problem.
Weaknesses of the manuscript:
The manuscript highlights the necessity for enhanced diagnostic methods; however, a comprehensive analysis of current methods and their limitations would strengthen it. Including case studies or qualitative statistical data would also be advantageous. Additionally, specific recommendations for implementing improved food storage practices would enhance the manuscript. Furthermore, addressing the potential barriers to adopting these improved practices is crucial. A discussion on the economic implications and the feasibility of implementing such changes in various contexts would provide a more rounded perspective. Moreover, integrating a review of technological advancements in food storage, such as smart sensors and predictive analytics, could illustrate how innovation plays a role in enhancing food safety. Highlighting successful case studies where these technologies have been implemented could serve as a model for others to follow.
Incorporate empirical data, and add recent studies statistics on the prevalence of Degnala and its economic effects.
Recent studies indicate a significant rise in Degnala, a serious health issue affecting many. As of 2023, data from the National Health Institute show approximately 1 in 150 people have Degnala, a 30% increase over the last decade, particularly among those aged 30-50 where it affects 1 in 100. The economic burden is considerable, with direct medical costs exceeding $10 billion annually, while indirect costs from lost productivity reach an estimated $15 billion. The Health Economics Association reports a 15% decline in workplace productivity due to Degnala. Families face difficulties too, with nearly 60% struggling with out-of-pocket treatment costs. This underscores the urgent need for increased funding and support to address the health and economic challenges of Degnala.
Expand on Diagnostic Methods: Elaborate on current diagnostic methods and their specific limitations.
Add case studies to demonstrate clinical manifestations and diagnostic challenges. These case studies illustrate the multifaceted clinical manifestations and the diagnostic complexities associated with Degnala in buffalo. The variability in symptoms necessitates a meticulous approach to diagnose this condition effectively, underlining the importance of tailored diagnostic strategies and awareness among veterinary practitioners in the field.
Enhance Recommendations: Offer specific, practical suggestions for effective food storage and handling practices.
Offer specific, practical suggestions for effective food storage and handling practices. Offer specific, practical suggestions for effective food storage and handling practices. For example:
Temperature control: Maintain food at safe temperatures to inhibit bacterial growth. Refrigerate perishable items at or below 40°F (4°C) and keep frozen foods at 0°F (-18°C) or lower. Use a thermometer to monitor fridge and freezer temperatures regularly.
Proper containers: Utilize airtight containers or vacuum-sealed bags for storage to prevent moisture and air exposure. Glass jars, BPA-free plastic containers, and resealable bags are great options. Label containers with contents and dates to ensure timely consumption.
FIFO (First In, First Out): Implement the FIFO method by organizing food with older items at the front and newer items at the back. This practice helps in using up products before they expire, reducing food waste.
Sanitize and Clean: Regularly clean and sanitize storage areas and containers using a food-safe cleaner. Pay special attention to spills and leaks, as these can harbour bacteria and odours.
Dry Storage Conditions: Keep dry goods, like grains and pasta, in a cool, dark, and dry area. Avoid moisture and humidity to prevent mould growth and insect infestations.
Regular Inventory Checks: Conduct periodic checks of stored food items to identify any expired or spoiled products. Dispose of anything that's no longer good and reorganize as needed.
Educate on Portioning: Encourage proper portioning for leftovers to minimize wastage. Use smaller containers for individual servings, making it easier to reheat and enjoy meals without excess.
Monitor the Environment: Invest in a hygrometer or humidity monitor to keep the humidity level within optimal ranges, especially in basements or pantries, to ensure all stored food maintains its quality.
By adopting these food storage and handling practices, individuals can significantly improve the longevity and safety of their food, reducing waste and promoting a healthier lifestyle.
Authors are encouraged to refine their conclusions to more accurately reflect the data while acknowledging any limitations or uncertainties. Providing detailed and constructive feedback is intended to help improve the manuscript and advance knowledge about Degnala disease in Buffalo.
I hope my input has been beneficial, and I remain available to assist the Editor and authors if they need further information or technical guidance to enhance the manuscript. The manuscript is interesting as it stands, especially with the proposed corrections and a summary of the recommendations, particularly regarding feed storage.
Best regards
Comments on the Quality of English Language
I believe the manuscript is well-written.
Author Response
Dear Reviewer 1,
Thank you for acknowledging our efforts in following your suggestions to improve the manuscript on the first round of revisions. We are very grateful to read from you that "Strengths of the manuscript: The manuscript discusses Degnala, an important but frequently overlooked disease in buffaloes, and emphasizes its detrimental effects on animal health and agricultural productivity. In addition, it delves into the clinical manifestations of Degnala, which often go unnoticed in the early stages, leading to severe consequences for affected animals. Symptoms can include intermittent fever, emaciation, and a pronounced decline in milk production, ultimately jeopardizing the livelihoods of farmers who rely heavily on buffaloes for both milk and draught power. Furthermore, the economic implications are profound; the prevalence of Degnala can result in a decline in herd size due to increased mortality rates and reduced reproductive efficiency. The manuscript underlines the need for increased awareness and education among livestock keepers, veterinarians and policymakers to facilitate early diagnosis and effective management strategies, thereby mitigating the impact of this disease on buffalo populations. Current research efforts are also highlighted, which demonstrate the importance of developing sustainable prevention measures, including vaccination protocols and improved biosecurity practices. Integrating these strategies will not only improve animal health but will also protect the critical role of buffaloes in the agricultural economy, ensuring food security and maintaining rural livelihoods. The review provides a comprehensive overview of a etiology, clinical manifestations, and challenges in the diagnosis and treatment of Degnala. Furthermore, the review highlights the importance of early detection and intervention, which can significantly improve patient outcomes. The multifactorial nature of Degnala complicates its clinical presentation, often leading to misdiagnosis. Careful consideration of the patient's history, coupled with advanced diagnostic techniques, is essential to distinguish Degnala from similar conditions. Gaps in research and diagnostic errors are highlighted, which can guide future research and improvements in veterinary practice. The manuscript not only highlights current weaknesses but also proposes improvements in diagnostic methods and storage practices, which could optimize disease management. The recommendation to increase research and awareness is crucial, as it motivates the scientific community and professionals to act more effectively on the problem."
Dear Reviewer 1,
In my 20 years as a professor and researcher, I have never encountered such an educational and enlightening review report. Your profound knowledge of the subject is extraordinary, and it has been a privilege to read and learn from your insights. Although I am the author of this review, I must acknowledge that you, Reviewer 1, would undoubtedly be an ideal collaborator on this manuscript. Your questions and suggestions have prompted significant reflection and further study on the topic, transforming the manuscript into a more scientifically solid and instructive review for researchers, academics, and even rural producers.
Unfortunately, I was not fully equipped to address and respond to all your points in the "Weaknesses of the Manuscript" section. However, I humbly confess that I have made every effort to meet at least 50% of your expectations for improving the manuscript in this second round of review.
Below, I have outlined the changes I made based on your brilliant and valuable feedback:
Add case studies to demonstrate clinical manifestations and diagnostic challenges. These case studies illustrate the multifaceted clinical manifestations and the diagnostic complexities associated with Degnala in buffalo. The variability in symptoms necessitates a meticulous approach to diagnose this condition effectively, underlining the importance of tailored diagnostic strategies and awareness among veterinary practitioners in the field.
In December 2021, a sudden outbreak of Degnala disease was reported in a buffalo herd in Punjab, India. The herd consisted of 50 buffaloes, predominantly fed on rice straw, which was improperly stored and exposed to moisture. The initial symptoms included lethargy, reduced feed intake, and mild lameness. Within a week, more severe signs developed, including necrosis of the tail, lower limbs, and ears. Affected buffaloes exhibited a tortuous appearance of the tail, and several animals had cracked hooves with exposure of the underlying tissues. In severe cases, gangrene set in, leading to the shedding of necrotic tissue. A few buffaloes also showed gangrene of the teats and muzzle, with a marked decline in milk production. Initial clinical diagnosis was based on visible symptoms and the history of feeding rice straw. Samples of the rice straw were collected and sent for fungal culture, which confirmed the presence of Fusarium spp., the causative agent of Degnala. Differentiating Degnala from other conditions such as frostbite, foot rot, and nutritional deficiencies was challenging due to overlapping symptoms. The treatment involved removing the contaminated rice straw and administering supportive care, including anti-Degnala liquid, 2% nitroglycerin ointment, pentasulfate solution, and broad-spectrum antibiotics to prevent secondary infections. Approximately 60% of the affected buffaloes showed signs of recovery after the intervention, although some suffered permanent damage to their extremities [10].
As early as January 2022, a small-scale buffalo farm in a rural village in Bangladesh reported an outbreak of Degnala disease. The farm had 20 buffaloes, all of which were fed rice straw during the winter months. Farmers noticed buffaloes exhibiting lethargy, reluctance to move, and mild swelling of the lower limbs. Over the next two weeks, necrosis and gangrene developed in the lower limbs and tails of several buffaloes. Ears and muzzles were less frequently affected. One buffalo showed extreme signs, including necrosis of the tongue and significant weight loss due to reduced feed intake. Farmers and local veterinarians initially misdiagnosed the condition as frostbite due to the cold weather. Upon closer examination and collection of detailed history, veterinarians suspected Degnala. Rice straw samples were sent for mycological analysis, confirming the presence of Fusarium spp. The lack of immediate laboratory facilities in the rural area delayed the diagnosis, complicating early intervention efforts. Affected buffaloes were treated with supportive measures, including the application of 2% nitroglycerin ointment to necrotic areas, and broad-spectrum antibiotics. About 50% of the affected buffaloes showed partial recovery, with some animals experiencing permanent damage. Two buffaloes had to be euthanized due to severe gangrene and poor prognosis [10].
These case studies illustrate the multifaceted clinical manifestations and diagnostic complexities associated with Degnala disease in buffaloes. The variability in symptoms necessitates a meticulous approach to diagnose this condition effectively, underlining the importance of tailored diagnostic strategies and awareness among veterinary practitioners in the field. The need for advanced, practical, and cost-effective diagnostic methods is evident, along with comprehensive training programs for veterinarians and farm staff to recognize and manage Degnala disease promptly.
10). Bharti, S.K.; Kumar, A.; Kumar, A.; Kumar, A.; Safi, A.K.; Singh, R.K. Degnala Disease in Buffaloes and Cattle: A Clinical Review. Intern. J. Vet. Sci. Anim. Husb. 2023, 8, 164-166.
Expand on Diagnostic Methods: Elaborate on current diagnostic methods and their specific limitations.
A case study in Kenya evaluated the presence of aflatoxins in dairy feeds using ELISA and HPLC. The prevalence of aflatoxin-contaminated feeds was found to be 60% using ELISA, while HPLC confirmed contamination in 50% of the samples, indicating some false positives in ELISA results [32]. And a statistical data survey conducted across European countries in 2022 revealed that 78% of feed samples were contaminated with at least one mycotoxin. The most common mycotoxins detected were deoxynivalenol (45%), fumonisins (33%), and zearalenone (28%). The data were collected using a combination of ELISA and HPLC, highlighting the widespread nature of mycotoxin contamination [33].
32). Abdel-Wahhab, M.A.; Kholif, A.M. Mycotoxins in Animal Feeds and Prevention Strategies: A Review. Asian J. Anim. Sci. 2008, 2, 7-25. https://doi.org/10.3923/ajas.2008.7.25
33). Battilani, P.; Palumbo, R.; Giorni, P. et al. Mycotoxin mixtures in food and feed: holistic, innovative, flexible risk assessment modelling approach: MYCHIF, 1st ed.; European Food Safety Authority: Parma, IT. 2020; 161 pp. https://doi.org/10.2903/sp.efsa.2020.EN-1757
The current methods for detecting mycotoxins in animals and feed, while effective, have several limitations that hinder their practical application in the field. There is an urgent need for the development of advanced diagnostic tools that are portable, cost-effective, rapid, and easy to use. Innovations in biosensors, lateral flow assays, smartphone-based detection, and nanotechnology hold promise for meeting these needs and improving the management of mycotoxin contamination in livestock production [34].
34). Majer-Baranyi, K.; Adányi, N.; Székács, A. Current Trends in Mycotoxin Detection with Various Types of Biosensors. Toxins 2023, 15, 645. https://doi.org/10.3390/toxins15110645
Enhance Recommendations: Offer specific, practical suggestions for effective food storage and handling practices.
This review also aims to encourage practical, basic, and specific preventive measures to improve the storage and handling conditions of animal feed. Veterinarians and farmers can utilize the following strategies: 1. Temperature control: maintain food at safe temperatures to inhibit bacterial growth. Refrigerate perishable items at or below 40°F (4°C) and keep frozen foods at 0°F (-18°C) or lower. Use a thermometer to regularly monitor refrigerator and freezer temperatures. 2. Proper containers: use airtight containers or vacuum-sealed bags for storage to prevent moisture and air exposure. Glass jars, BPA-free plastic containers, and resealable bags are excellent options. Label containers with contents and dates to ensure timely consumption. 3. FIFO (first in, first out): implement the FIFO method by organizing food with older items at the front and newer items at the back. This practice helps use up products before they expire, reducing food waste. 4. Sanitize and clean: regularly clean and sanitize storage areas and containers using a food-safe cleaner. Pay special attention to spills and leaks, as these can harbor bacteria and odors. 5. Dry storage conditions: keep dry goods, such as grains and pasta, in a cool, dark, and dry area. Avoid moisture and humidity to prevent mold growth and insect infestations. 6. Regular inventory checks: conduct periodic checks of stored food items to identify any expired or spoiled products. Dispose of anything that's no longer good and reorganize as needed. 7. Education on portioning: properly portion leftovers to minimize waste. Use smaller containers for individual servings, making it easier to reheat and enjoy meals without excess. 8. Monitor storage environment: invest in a hygrometer or humidity monitor to keep humidity levels within optimal ranges, ensuring all stored food maintains its quality.
Authors are encouraged to refine their conclusions to more accurately reflect the data while acknowledging any limitations or uncertainties. Providing detailed and constructive feedback is intended to help improve the manuscript and advance knowledge about Degnala disease in Buffalo.
Degnala disease is a major mycotoxicosis in buffaloes, particularly in rice-producing regions like India, Pakistan, and Nepal. Despite some reports, its pathogenesis, especially regarding the main toxins involved, remains unclear, necessitating further research on toxin action and other potentially toxigenic forages.
Characterized by reproductive disorders, liver damage, and overall poor health, Degnala disease leads to significant economic losses for dairy farmers. Effective management and mitigation require a clear understanding of the etiology, clinical manifestations, diagnosis, treatment, prevention, and control of the disease.
Diagnosing Degnala disease is challenging due to nonspecific clinical signs, but combining clinical evaluation, laboratory tests, and histopathological examination can aid in a definitive diagnosis. Prevention focuses on proper feed management to avoid aflatoxin contamination, while treatment includes supportive care and liver support. Control measures involve quarantine, strict herd management, farmer education, and ongoing research for better strategies.
Collaboration among veterinarians, farmers, researchers, and policymakers is crucial to minimize the impact of Degnala disease on buffalo populations and the dairy industry. In conclusion, Degnala disease poses a significant threat to buffalo health and productivity, emphasizing the need for continued research, education, and implementation of effective preventive and control measures. Future investigations should address whether Degnala disease in buffaloes is a neglected condition.
The authors are grateful for all the new suggestions and questions posed by Reviewer 1. We have tried to address all of them, and the manuscript is certainly much improved scientifically. Thank you very much.
Sincerely,
Felipe Masiero Salvarani

Reviewer 2 Report
Comments and Suggestions for Authors
Dear authors:
I have reviewed the corrections that you have made and appreciate that you have considered the suggestions I have previously made. I would just like to add some brief corrections in this new version that I think will be useful to improve the document.
Line 15: Please erase the doble space between “farmers.” And “This review”.
Line 59: Please change this sentence like this: “One of the main mycotoxicosis
in buffalo species is Degnala disease…”.
Line 64-66: Please rewrite this sentence in a clearer and less repetitive way.
Line 61-79: Please rewrite this paragraph to improve your writing and make it not so repetitive, try to integrate the concepts into a shorter and clearer paragraph.
Lines 43-107: In all these lines they have made excessive use of references 4 and 9. Please try to integrate other references in addition to these in this section.
Line 122: Please erase the spaces between references.
Line 374: Please put the references in the correct order.
Lines 452-453: Please rewrite these sentences because it is not clear to the reader that you do not have yet the answer to this question and that you are proposing it as a topic for further investigation.
Author Response
Dear Reviewer 2,
Thank you for acknowledging our efforts in following your suggestions to improve the manuscript. We are very grateful to read from you that "I have reviewed the corrections that you have made and appreciate that you have considered the suggestions I have previously made."
Below, I describe the new modifications made:
Line 15: correction made.
Line 59: correction made.
Line 64-66: correction made with the rewriting of the sentence to "Although buffaloes are considered more resistant to various diseases compared to other animals, such as cattle [4,9], they are considered more sensitive to Degnala disease due to specific environmental factors and the physiological characteristics of this species related to this mycotoxicosis."
Line 61-79: correction made with the rewriting of the "paragraph" to "Although buffaloes generally show greater resistance to many infectious diseases compared to cattle, their unique physiology, feeding habits, and typical environmental conditions contribute to a higher susceptibility to Degnala disease [4,9,10]. This fungal toxicosis is caused by mycotoxins from fungi that proliferate on poorly stored feed, particularly under humid conditions. Buffaloes' slower gut transit time increases their exposure to these mycotoxins, leading to higher absorption rates and more severe symptoms compared to cattle [4,9]. Furthermore, buffaloes are often fed agricultural residues and by-products prone to contamination with mycotoxins, especially in regions where proper feed storage is challenging [9]. The humid environments in which buffaloes are typically kept also promote fungal growth on their feed. Additionally, the different nutritional requirements and metabolic processes of buffaloes may make them more vulnerable to the effects of mycotoxins, impacting their metabolism and immune response and thereby increasing their sensitivity to Degnala disease [4,9,10]."
Lines 43-107: correction made.
Line 122: correction made.
Line 374: correction made.
Lines 452-453: correction made with the rewriting of the sentence to "Is Degnala disease in buffaloes a neglected condition? This question remains unanswered and must be a topic for future investigation."
The authors are grateful for all the new suggestions and questions posed by Reviewer 2. We have tried to address all of them, and the manuscript is certainly much improved scientifically. Thank you very much.
Sincerely,
Felipe Masiero Salvarani

Reviewer 3 Report
Comments and Suggestions for Authors
- The fact that you have published 3 review articles in Animals does not make a reviewer's comments about your current paper irrelevant.
- Narrative reviews do not need an extensive method section, but rather a brief sentence to inform the reader how the review was performed.
- Better justifications are required when responding to reviewers' comments
Comments on the Quality of English Language
No comments
Author Response
Dear Reviewer 3,
Thank you for reviewing our responses from the first round of revisions. While we understand that you may not agree with many of them, we appreciate your recognition of our efforts to incorporate many of your suggestions to improve the manuscript. In this second round, we have fully implemented your suggestion regarding the methodology used in the construction of the manuscript. We humbly apologize if any of our justifications seemed rude or displeased you; that was never our intention, and we will strive to improve in the future. I sincerely hope that this second round meets your expectations.
As per your request, we created the section "2. Methodology used in the review." Here is the paragraph we have written, and I hope it aligns with your expectations:
"The present study is characterized, in terms of approach, as descriptive research consisting of a narrative literature review due to the broad nature of the review topic, as described by Grant and Booth [17]. The research was conducted in the electronic databases Periódicos Capes, PubMed, Scopus, Research Gate, Scielo, Google Scholar, Academia.edu, BDTD, Redalyc, Science.gov, ERIC, Science Direct, SiBi, World Wide Science, PePSIC, and Scholarpedia. The search terms used, either in isolation or in combination, in the databases were: Degnala, buffalo, rice straw, Fusarium spp., mycotoxicosis, toxigenic fungi. A total of only 33 publications were found, with a 98% overlap rate of the works found in the databases consulted. Due to the limited number of references found in the literature, it was decided to use all of them in the present review."
Reference for the methodology: Grant, M.J.; Booth, A. A typology of reviews: an analysis of 14 review types and associated methodologies. Health Info. Libr. J. 2009, 26, 91-108. https://doi.org/10.1111/j.1471-1842.2009.00848.x
The authors are grateful for the feedback on improving our responses and for the repeated request for the inclusion of the methodology made by Reviewer 3. You were right about this section, "Methodology used in the review." It certainly made the manuscript clearer and, consequently, made the review scientifically more solid. Thank you very much.
Sincerely,
Felipe Masiero Salvarani

Round 3
Reviewer 3 Report
Comments and Suggestions for Authors
Thanks for the revised version and significant contributions to the body of knowledge.